# Prediction with Corrupted Expert Advice

**Idan Amir**[*]
Tel Aviv University
idanamir@mail.tau.ac.il

**Idan Attias**[*]
Ben-Gurion University
idanatti@post.bgu.ac.il

**Tomer Koren**
Tel Aviv University and Google
tkoren@tauex.tau.ac.il

**Roi Livni**
Tel Aviv University
rlivni@tauex.tau.ac.il

**Yishay Mansour**
Tel Aviv University and Google
mansour.yishay@gmail.com

## Abstract

We revisit the fundamental problem of prediction with expert advice, in a setting where the environment is benign and generates losses stochastically, but the feedback observed by the learner is subject to a moderate adversarial corruption. We prove that a variant of the classical Multiplicative Weights algorithm with decreasing step sizes achieves constant regret in this setting and performs optimally in a wide range of environments, regardless of the magnitude of the injected corruption. Our results reveal a surprising disparity between the often comparable Follow the Regularized Leader (FTRL) and Online Mirror Descent (OMD) frameworks: we show that for experts in the corrupted stochastic regime, the regret performance of OMD is in fact strictly inferior to that of FTRL.

## 1 Introduction

Prediction with expert advice is perhaps the single most fundamental problem in online learning and sequential decision making. In this problem, the goal of a learner is to aggregate decisions from multiple experts and achieve performance that approaches that of the best individual expert in hindsight. The standard performance criterion is the regret: the difference between the loss of the learner and that of the best single expert. The experts problem is often considered in the so-called adversarial setting, where the losses of the individual experts may be virtually arbitrary and even be chosen by an adversary so as to maximize the learner's regret. The canonical algorithm in this setup is the Multiplicative Weights algorithm [21, 12], that guarantees an optimal regret of $\Theta(\sqrt{T \log N})$ in any problem with $N$ experts and $T$ decision rounds.

A long line of research in online learning has focused on obtaining better regret guarantees, often referred to as "fast rates," on benign problem instances in which the loss generation process behaves more favourably than in a fully adversarial setup. A prototypical example of such an instance is the stochastic setting of the experts problem, where the losses of the experts are drawn i.i.d. over time from a fixed and unknown distribution, and there is a constant gap $\Delta$ between the mean losses of the best and second-best experts. In this setting, it has been established that the optimal expected regret scales as $\Theta(\log(N)/\Delta)$, and in particular, is bounded by a constant independent of the number of rounds $T$ [7, 20]. More recently, [25] have shown that this optimal regret is in fact achieved by an

---

[*]Equal contribution.

adaptive variant of the multiplicative weights algorithm. Other works have studied various intermediate regimes between stochastic and adversarial, where the challenge is to adapt to the complexity of the problem with little or no prior knowledge (e.g., [5, 15, 6, 28, 19, 29, 18, 32, 9, 31, 10, 11]).

In this work, we consider a different, natural intermediate regime of the experts problem: an adversarially-corrupted stochastic setting. Here, the adversary can modify the stochastic losses with arbitrary corruptions, as long as the sum of the corruptions is bounded by a parameter $C$, which is unknown to the learner. One application domain where corruptions are natural is content/ads recommendation: the presence of malicious users affects the feedback signal received by the learning algorithm, but the objective one cares about is the performance of the system (measured via pseudo regret) on the true population of non-malicious users. The injection of adversarial corruptions implies that the learner observes losses which are *not* distributed i.i.d. across time steps. In principle, one could use the adversarial online learning approach to overcome this challenge, but this will result in significantly inferior regret bounds that scale polynomially with the time horizon. The challenge is then to extend the favourable constant bounds on the regret achievable in the purely stochastic setting to allow for moderate adversarial corruptions.

In the closely related Multi-Armed Bandit (MAB) partial-information model in online learning, the adversarially-corrupted stochastic setting has recently received considerable attention [23, 13, 34, 16, 17, 22]. (Even more recently, a similar setting was also considered in episodic reinforcement learning [24].) Yet, the natural question of determining the optimal regret rate in the analogous full-information problem remained open. Given that the optimal bounds in the bandit setting scale linearly with the number of experts (or "arms" in the context of MAB), it becomes a fundamental question if this dependence can be reduced to logarithmic with full-information, while preserving the dependence on the other parameters of the problem.

Indeed, our main result shows that the optimal regret in the adversarially-corrupted stochastic setting scales as $\Theta(\log(N)/\Delta + C)$ independently of the horizon $T$, and moreover, this optimal bound is attained by a simple adaptive variant of the classic multiplicative weights algorithm, that does not require knowing the corruption level $C$ in advance. In fact, it turns out that this simple algorithm performs optimally in all three regimes simultaneously: the pure stochastic setting, the adversarially-corrupted setting, and the fully-adversarial setting. It is important to note that this kind of behaviour is *not* an immediate consequence of the known $O(\log(N)/\Delta)$ performance in the stochastic case, as presented in [25]. Even a small amount of adversarial corruption might hinder the algorithm from rapidly concentrating its decisions on the best (uncorrupted) expert, and in principle, this could potentially have a longer-term effect than just on the $C$ corrupted rounds themselves.

Our strategy for proving these results is based on a novel and delicate analysis of the adaptive multiplicative weights algorithm in the stochastic case, which can be seen as analogous to the approach taken by [33, 34] in multi-armed bandits. The first step in this analysis adapts a standard worst-case regret bound for multiplicative weights with an explicit dependence on the second-moments of the losses to the case of an adaptive step-size sequence. Then, we observe that the second-order terms admit a "self-bounding" property and their sum can be bounded by the (pseudo-)regret itself. The other expression in the regret bound, which is a sum of entropy terms that stems from the changing step sizes and captures the stability of the algorithm, is more challenging to handle; we show that this sum is also self-bounded by the regret up to exponentially-decreasing terms that sum up to a constant. Putting these together lead to a constant regret bound in the stochastic case. Crucially, since the said arguments are all inherently worst-case and do not directly rely on the i.i.d. nature of the losses, the whole analysis turns out to be robust to corruptions and yields the additive $C$ term in the moderately corrupted case.

An interesting byproduct of our analysis is a surprising disparity between two common online learning meta-algorithms: Follow the Regularized Leader (FTRL) and Online Mirror Descent (OMD). We show that while both FTRL and OMD give rise to optimal (adaptive) multiplicative weights algorithms in the pure stochastic experts setting,[2] the OMD variant becomes strictly inferior to the FTRL variant once corruptions are introduced, and has a much weaker regret of $\Omega(C/\Delta)$ for a fixed number of experts $N$. In contrast, the non-adaptive (i.e., fixed step size) variants of the meta-algorithms are well-known to be equivalent in the more general setting of online linear optimization. We note that a closely related separation result was shown by [27] in the standard adversarial setup, who

demonstrated a case where OMD suffers linear regret whereas FTRL guarantees a $\sqrt{T}$-type bound. Here, we give a specialized argument in the moderately corrupted setting that reveals a more intricate dependence on the complexity of the problem, in terms of the parameters $C$ and $\Delta$. We also show a few basic simulations in which this gap is clearly visible and tightly supports our theoretical bounds.

## 2 Preliminaries

### 2.1 Problem setup

We consider the classic problem of prediction with expert advice, with a set of $N$ experts indexed by $[N] = \{1, \ldots, N\}$. In each time step $t = 1, \ldots, T$ the learner chooses a probability vector $p_t = (p_{t,1}, \ldots, p_{t,N})$ from the simplex $\mathcal{S}_N = \{p \in \mathbb{R}^N : \forall i, \ p_i \geq 0 \text{ and } \sum_{i=1}^{N} p_i = 1\}$. Thereafter, a loss vector $\ell_t \in [0,1]^N$ is revealed. We will consider three variants of the problem, as follows.

In the *adversarial* (non-stochastic) setting, the loss vectors $\ell_1, \ldots, \ell_T$ are entirely arbitrary and may be chosen by an adversary. The goal of the learner is to minimize the regret, given by

$$\mathcal{R}_T := \sum_{t=1}^{T} p_t \cdot \ell_t - \min_{i \in [N]} \sum_{t=1}^{T} \ell_{t,i}.$$

In the *stochastic* setting, the loss vectors $\ell_1, \ldots, \ell_T$ are drawn i.i.d. from a fixed (and unknown) distribution. We denote the vector of the mean losses by $\mathbf{E}[\ell_t] = \mu = (\mu_1, \ldots, \mu_N)$, and let $i^\star = \operatorname{argmin}_{i \in [N]} \mu_i$ be the index of the best expert, which we assume is unique. The gap between any expert $i$ and best one is denoted $\Delta_i = \mu_i - \mu_{i^\star}$, and we let $\Delta = min_{i \neq i^\star}\{\mu_i - \mu_{i^\star}\} > 0$. The goal of the learner in the stochastic setting is to minimize the *pseudo regret*, defined as

$$\overline{\mathcal{R}}_T := \sum_{t=1}^{T} p_t \cdot \mu - \sum_{t=1}^{T} \mu_{i^\star} = \sum_{t=1}^{T} \sum_{i=1}^{N} p_{t,i} \left( \mu_i - \mu_{i^\star} \right). \tag{1}$$

Finally, in the *adversarially-corrupted stochastic* setting (following [23, 13]), which is the main focus of this paper, loss vectors $\ell_1, \ldots, \ell_T$ are drawn i.i.d. from a fixed and unknown distribution as in the stochastic setting with mean rewards $\mu = \mathbf{E}[\ell_t]$, and the same definitions of best expert $i^\star$ and gap $\Delta$. Subsequently, an adversary is allowed to manipulate the feedback observed by the learner, up to some budget $C > 0$ which we refer to as the corruption level. Formally, on each round $t = 1, \ldots, T$:

(1) A stochastic loss vector $\ell_t \in [0,1]^N$ is drawn i.i.d. from a fixed and unknown distribution;
(2) The adversary observes the loss vector $\ell_t$ and generates corrupted losses $\tilde{\ell}_t \in [0,1]^N$;
(3) The player picks a distribution $p_t \in \mathcal{S}_N$ over experts, suffers the loss $p_t \cdot \ell_t$, and observes *only the corrupted* loss vector $\tilde{\ell}_t$.

Notice that we allow the adversary to be fully adaptive, in the sense that the corruption on round $t$ may depend on past choices of the learner (before round $t$) as well as on the realizations of the random loss vectors $\ell_1, \ldots, \ell_t$ in all rounds up to (and including) round $t$.

We consider the following measure of corruption, which we assume to be unknown to the learner:

$$C = \sum_{t=1}^{T} \|\tilde{\ell}_t - \ell_t\|_\infty. \tag{2}$$

Like in the stochastic setting, the goal of the learner is to minimize the *pseudo regret* (defined in Eq. (1)). Note that, crucially, the pseudo regret of the learner depends only on the (means of) the stochastic losses $\ell_t$ and the adversarial corruption appears only in the feedback observed by the learner.

### 2.2 Multiplicative Weights

We recall two variants of the classic Multiplicative Weights (MW) algorithm that we revisit in this work. The standard MW algorithm [21, 12] is parameterized by a fixed step-size parameter $\eta > 0$.

For an arbitrary sequence of loss vectors $g_1, \ldots, g_T \in \mathbb{R}^N$, it admits the following update rule, on every round $t$:

$$p_{t,i} = \frac{e^{-\eta \sum_{s=1}^{t-1} g_{s,i}}}{\sum_{j=1}^{N} e^{-\eta \sum_{s=1}^{t-1} g_{s,j}}}, \qquad \forall\, i \in [N]. \tag{3}$$

For the basic, fixed step-size version of our results, we will need a standard second-order regret bound for MW.

**Lemma 1** ([5]; see also [2])**.** *If $|g_{t,i}| \leq 1$ for all $t \geq 1$ and $i \in [N]$, the regret of the MW updates in Eq. (3) is bounded as*

$$\sum_{t=1}^{T} \sum_{i=1}^{N} p_{t,i}\big(g_{t,i} - g_{t,i^\star}\big) \leq \frac{\log N}{\eta} + \eta \sum_{t=1}^{T} \sum_{i=1}^{N} p_{t,i} g_{t,i}^2.$$

In particular, the bound implies the well-known $\Theta(\sqrt{T \log N})$ optimal regret bound for MW in the adversarial setting, if the step size is properly tuned to $\eta = \Theta(\sqrt{\log(N)/T})$; note that this setting of $\eta$ depends on the time horizon $T$.

An adaptive variant of the MW algorithm that does not require knowledge of $T$ was proposed in [3]. This variant employs a diminishing step size sequence, and takes the form:

$$p_{t,i} = \frac{e^{-\eta_t \sum_{s=1}^{t-1} g_{s,i}}}{\sum_{j=1}^{N} e^{-\eta_t \sum_{s=1}^{t-1} g_{s,i}}}, \qquad \forall\, i \in [N], \tag{4}$$

with $\eta_t = \sqrt{\log(N)/t}$ for all $t \geq 1$. This algorithm was shown to obtain the optimal $\Theta(\sqrt{T \log N})$ regret in the adversarial setup for any $T$ [3, 4]. We will show that, remarkably, the adaptive MW algorithm also achieves the optimal performance in the adversarially-corrupted experts setting, for any level of corruption.

We remark that the MW algorithm in Eq. (4) is in fact an instantiation of the canonical Follow-the-Regularized Leader (FTRL) framework in online optimization with entropy as regularization, when one allows the magnitude of regularization to change from round to round. MW can also be obtained by instantiating the closely related Online Mirror Descent (OMD) meta-algorithm, that also allows for the regularization to vary across rounds. (For more background on online optimization, FTRL and OMD, see the full version of the paper [1].) When the regularization is fixed, it is a well-known fact that the two frameworks are generically equivalent and give rise to precisely the same algorithm, presented in Eq. (3). However, when the regularization is time-dependent, they produce different algorithms. We discuss the disparities between these different variants in more details in Section 3.3.

## 3   Main Results

In this section, we consider the adversarially-corrupted stochastic setting and present our main results. As a warm-up, we analyze the Multiplicative Weights algorithm with fixed step sizes while assuming the minimal gap $\Delta$ is known to the learner. Then, we consider the general case where neither the gap $\Delta$ nor the corruption level $C$ are known, and prove that the adaptive multiplicative weights algorithm attains optimal performance.

### 3.1   A warm-up analysis for known minimal gap

We begin with an easier case where the gap $\Delta$ is known to the learner, and can be used to tune the step size parameter of multiplicative weights (Eq. (3)). In this case, a fixed step-size algorithm suffices and we have the following.

**Theorem 2.** *The Multiplicative Weights algorithm (Eq. (3)) with $\eta = \Delta/2$ in the adversarially-corrupted stochastic regime with corruption level $C$ over $T$ rounds, achieves constant $O(\log(N)/\Delta + C)$ expected pseudo regret.*

Two basic observations in the analysis are the following. The first observation gives a straightforward bound on the *corrupted* losses of an expert in terms of its pseudo regret.

**Observation 3.** *For any $t = 1, \ldots, T$ and $i \in [N]$ the following holds*

$$\left(\tilde{\ell}_{t,i} - \tilde{\ell}_{t,i^\star}\right)^2 \leq \frac{1}{\Delta}\left(\mu_i - \mu_{i^\star}\right).$$

**Proof.** For $i \neq i^\star$, note that $\frac{1}{\Delta}(\mu_i - \mu_i^\star) \geq 1$, and on the other hand, $(\tilde{\ell}_{t,i} - \tilde{\ell}_{t,i^\star})^2 \leq 1$ since $\tilde{\ell}_{t,i} \in [0,1]$. On the other hand, for $i = i^\star$ we have $\frac{1}{\Delta}(\mu_i - \mu_i^\star) = 0$ and $(\tilde{\ell}_{t,i} - \tilde{\ell}_{t,i^\star})^2 = 0$. ∎

The second observation relates the regret with respect to the corrupted and uncorrupted losses.

**Observation 4.** *For any probability vectors $\{p_t \in \mathcal{S}_N : t = 1 \ldots T\}$ the following holds*

$$\sum_{t=1}^{T} \sum_{i=1}^{N} p_{t,i}\left(\ell_{t,i} - \ell_{t,i^\star}\right) \leq \sum_{t=1}^{T} \sum_{i=1}^{N} p_{t,i}\left(\tilde{\ell}_{t,i} - \tilde{\ell}_{t,i^\star}\right) + 2C.$$

**Proof.** Denoting $\delta_{t,i}$ as the corruption for expert $i$ at time step $t$, we get

$$\sum_{t=1}^{T} \sum_{i=1}^{N} p_{t,i}\left(\tilde{\ell}_{t,i} - \tilde{\ell}_{t,i^\star}\right) = \sum_{t=1}^{T} \sum_{i=1}^{N} p_{t,i}\left(\ell_{t,i} - \ell_{t,i^\star}\right) + \sum_{t=1}^{T} \sum_{i=1}^{N} p_{t,i}\left(\delta_{t,i} - \delta_{t,i^\star}\right).$$

By definition of the corruption $\sum_{t=1}^{T} \max_{i \in [N]} |\delta_{t,i}| \leq C$ and therefore $\sum_{t=1}^{T} \sum_{i=1}^{N} p_{t,i}|\delta_{t,i}| \leq C$. Using the triangle inequality implies that $\sum_{t=1}^{T} \sum_{i=1}^{N} p_{t,i}(\delta_{t,i} - \delta_{t,i^\star}) \geq -2C$. ∎

We now turn to prove the theorem.

**Proof of Theorem 2.** We start off with the basic bound of (fixed step size) MW in Lemma 1:

$$\sum_{t=1}^{T} \sum_{i=1}^{N} p_{t,i}\left(\tilde{\ell}_{t,i} - \tilde{\ell}_{t,i^\star}\right) \leq \frac{\log N}{\eta} + \eta \sum_{t=1}^{T} \sum_{i=1}^{N} p_{t,i}\tilde{\ell}_{t,i}^2.$$

First, note that the regret of playing a fixed sequence $p_1, \ldots, p_T$ is not affected by an additive translation of the form $\tilde{\ell}'_{t,i} = \tilde{\ell}_{t,i} - a'_t$ for any constant $a_t$ such that $\tilde{\ell}'_{t,i} \in [-1,1]$. In addition, for the Multiplicative Weights algorithm the sequences $p_1, \ldots, p_T$ are also not affected by additive translation. Thus, taking $a_t = \tilde{\ell}_{t,i^\star}$ yields

$$\sum_{t=1}^{T} \sum_{i=1}^{N} p_{t,i}\left(\tilde{\ell}_{t,i} - \tilde{\ell}_{t,i^\star}\right) \leq \frac{\log N}{\eta} + \eta \sum_{t=1}^{T} \sum_{i=1}^{N} p_{t,i}\left(\tilde{\ell}_{t,i} - \tilde{\ell}_{t,i^\star}\right)^2.$$

Applying observations 3 and 4 and rearranging terms implies

$$\sum_{t=1}^{T} \sum_{i=1}^{N} p_{t,i}\left(\ell_{t,i} - \ell_{t,i^\star}\right) \leq \frac{\log N}{\eta} + 2C + \frac{\eta}{\Delta} \sum_{t=1}^{T} \sum_{i=1}^{N} p_{t,i}\left(\mu_i - \mu_{i^\star}\right).$$

Taking expectation while using the fact that $p_{t,i}$ and $\ell_{t,i}$ are independent we obtain

$$\mathbf{E}\left[\overline{\mathcal{R}}_T\right] \leq \frac{\log N}{\eta} + 2C + \frac{\eta}{\Delta}\mathbf{E}\left[\overline{\mathcal{R}}_T\right].$$

Finally, by setting $\eta = \Delta/2$ and rearranging we can conclude that

$$\mathbf{E}[\overline{\mathcal{R}}_T] \leq \frac{4\log N}{\Delta} + 4C.$$ ∎

## 3.2 General analysis with decreasing step sizes

We now formally state and prove our main result: a constant regret bound in the adversarially-corrupted case for the adaptive MW algorithm (in Eq. (4)), that does not require the learner to know neither the gap $\Delta$ nor the corruption level $C$.

**Theorem 5.** *The adaptive MW algorithm in Eq. (4) with $\eta_t = \sqrt{\log(N)/t}$ in the adversarially-corrupted stochastic regime with corruption level $C$ over $T$ rounds, achieves constant $O(\log(N)/\Delta + C)$ expected pseudo regret.*

Note that this result is tight (up to constants): a lower bound of $\Omega(\log(N)/\Delta)$ was shown by [25], and a lower bound of $\Omega(C)$ is straighforward: consider an instance with $N = 2$ experts, means 0 and 1 (assigned randomly to the experts) and an adversary that corrupts the first $C$ rounds and assigns a loss of zero to both experts on those rounds; the learner receives no information about the identity of the best expert (whose mean loss is the smallest) during the first $C$ rounds and thus incurs, in expectation, at least $C/2$ pseudo regret over these rounds.

For the proof of Theorem 5 we require two main lemmas. The first lemma is a second-order regret bound for adaptive MW, analogous to the one stated in Lemma 1 for the fixed step size case. Here and throughout the section, we use $H(\cdot)$ to denote the entropy of a probability vector, that is, $H(p) = \sum_{i=1}^{N} p_i \log(1/p_i)$.

**Lemma 6.** *For any sequence of loss vectors $g_1, \ldots, g_T \in \mathbb{R}^N$, the regret of the adaptive MW algorithm in Eq. (4) satisfies*

$$\sum_{t=1}^{T} \sum_{i=1}^{N} p_{t,i}(g_{t,i} - g_{t,i^\star}) \leq 4 \log N + \frac{1}{2 \log N} \sum_{t=1}^{T} \eta_t H(p_{t+1}) + 5 \sum_{t=1}^{T} \eta_t \sum_{i=1}^{N} p_{t,i} g_{t,i}^2,$$

*provided that $|g_{t,i}| \leq 1$ for all $t$ and $i \in [N]$.*

The lemma is obtained from a more general bound for Follow-the-Regularized Leader, and follows from standard arguments adapted to the case of time-varying regularization. For completeness, we give this derivation in the full version of the paper [1]. The second lemma, key to our refined analysis of adaptive MW, shows that a properly scaled version of the entropy of any probability vector $p$ is upper bounded by the instantaneous pseudo regret of $p$, up to an exponentially decaying additive term.

**Lemma 7.** *For any $N > 0$, $0 < \Delta \leq 1$ and $\tau \geq \tau_0 = 64 \Delta^{-2} \log^2 N$, we have the following bound for the entropy of any probability vector $p$ and any $i^\star \in [N]$:*

$$\frac{1}{\sqrt{\tau}} H(p) \leq \frac{5}{8} \sum_{i \neq i^\star} p_i \Delta + \frac{2}{\sqrt{\tau}} e^{-\frac{1}{8} \Delta \sqrt{\tau}}.$$

We prove the lemma below, but first let us show how it is used to derive our main theorem.

**Proof of Theorem 5.** Applying Lemma 6 on the corrupted loss vectors $g_t = \tilde{\ell}_t$ and introducing additive translations of $\tilde{\ell}_{t,i^\star}$ as before, yields the bound

$$\sum_{t=1}^{T} \sum_{i=1}^{N} p_{t,i}(\tilde{\ell}_{t,i} - \tilde{\ell}_{t,i^\star}) \leq 4 \log N + \frac{1}{2 \log N} \sum_{t=1}^{T} \eta_t H(p_{t+1}) + 5 \sum_{t=1}^{T} \eta_t \sum_{i=1}^{N} p_{t,i}(\tilde{\ell}_{t,i} - \tilde{\ell}_{t,i^\star})^2.$$

In Lemma 12 (see the full version of the paper [1]) we bound the last term in the bound in terms of the pseudo regret (similarly to the proof of Theorem 2), as follows:

$$\sum_{t=1}^{T} \eta_t \sum_{i=1}^{N} p_{t,i}(\tilde{\ell}_{t,i} - \tilde{\ell}_{t,i^\star})^2 \leq \frac{16 \log N}{\Delta} + \frac{1}{8} \overline{\mathcal{R}}_T.$$

For bounding the first summation in the bound, we use Lemma 7. Summing the lemma's bound over $t = 1, \ldots, T$ and bounding the sum of the exponential terms by an integral (refer to Lemma 13 in the full version of the paper [1] for the details), we obtain

$$\frac{1}{\log N} \sum_{t=1}^{T} \eta_t H(p_{t+1}) \leq \frac{50 \log N}{\Delta} + \frac{5}{8} \overline{\mathcal{R}}_T.$$

Plugging the two inequalities into the regret bound, we obtain

$$\sum_{t=1}^{T} \sum_{i=1}^{N} p_{t,i}(\tilde{\ell}_{t,i} - \tilde{\ell}_{t,i^\star}) \leq \frac{109 \log N}{\Delta} + \frac{15}{16} \overline{\mathcal{R}}_T.$$

Using observation 4 and taking expectation we get

$$\mathbf{E}[\overline{\mathcal{R}}_T] \leq \frac{109 \log N}{\Delta} + 2C + \frac{15}{16} \mathbf{E}[\overline{\mathcal{R}}_T].$$

Rearranging terms gives the theorem. ∎

We conclude this section with a proof of our key lemma.

**Proof of Lemma 7.** We split the analysis of the sum for $i \neq i^\star$ and $i = i^\star$. Considering first the case $i = i^\star$, we apply the inequality $\log x \leq x - 1$ for $x \geq 1$ to obtain, for $\tau \geq \tau_0 \geq 64/\Delta^2$,

$$\frac{1}{\sqrt{\tau}} p_{i^\star} \log \frac{1}{p_{i^\star}} \leq \frac{1}{\sqrt{\tau}}(1 - p_{i^\star}) \leq \frac{1}{8} \sum_{i \neq i^\star} p_i \Delta.$$

Next, we examine the remaining terms with $i \neq i^\star$. The main idea is to look at two different regimes: one when $p_i > e^{-\frac{1}{2}\Delta\sqrt{\tau}}$ and the other for $p_i \leq e^{-\frac{1}{2}\Delta\sqrt{\tau}}$. In the former case, we have

$$\frac{1}{\sqrt{\tau}} p_i \log \frac{1}{p_i} \leq \frac{1}{2\sqrt{\tau}} p_i \Delta\sqrt{\tau} = \frac{1}{2} p_i \Delta.$$

For the latter case, we can use the inequality of $\log x \leq 2\sqrt{x}$ for $x > 0$ to obtain

$$\frac{1}{\sqrt{\tau}} p_i \log \frac{1}{p_i} \leq \frac{2}{\sqrt{\tau}} \sqrt{p_i} \leq \frac{2}{\sqrt{\tau}} e^{-\frac{1}{4}\Delta\sqrt{\tau}}.$$

Combining both observations for $i \neq i^\star$ implies

$$\frac{1}{\sqrt{\tau}} \sum_{i \neq i^\star} p_i \log \frac{1}{p_i} \leq \frac{1}{2} \sum_{i \neq i^\star} p_i \Delta + \frac{2N}{\sqrt{\tau}} e^{-\frac{1}{4}\Delta\sqrt{\tau}}.$$

Finally, note that for $\tau \geq \tau_0 = 64 \log^2(N)/\Delta^2$ it holds that $e^{-\frac{1}{4}\Delta\sqrt{\tau}} \leq e^{-\frac{1}{8}\Delta\sqrt{\tau} - \log N} = N^{-1} e^{-\frac{1}{8}\Delta\sqrt{\tau}}$. This together with our first inequality concludes the proof. ∎

### 3.3  Gap between Follow the Regularized Leader and Online Mirror Descent

Here we present a surprising contrast between the variants of the adaptive MW algorithm obtained by instantiating the Follow the Regularized Leader (FTRL) and Online Mirror Descent (OMD) meta-algorithms, in the adversarially corrupted regime. We show that while both give optimal algorithms in the stochastic experts setting, the OMD variant becomes strictly inferior to the FTRL variant once corruptions are introduced.

As remarked above, when the step size (i.e., the magnitude of regularization) is fixed, the two meta-algorithms are equivalent, and produce the classic MW algorithm in Eq. (3) when their regularization is set to the negative entropy function over the probability simplex. (For more background and references, see the full version of the paper [1].) Once one allows the step-sizes $\eta_t$ to vary across rounds, the FTRL gives the adaptive MW algorithm in Eq. (4), while OMD yields the following updates:

$$p_{t,i} = \frac{e^{-\sum_{s=1}^{t-1} \eta_s \ell_{s,i}}}{\sum_{j=1}^{N} e^{-\sum_{s=1}^{t-1} \eta_s \ell_{s,i}}}, \qquad \forall i \in [N]. \tag{5}$$

First, we show that the OMD variant of MW in Eq. (5) obtains the same constant $O(\log(N)/\Delta)$ regret bound in the pure stochastic regime, up to small $\log \log N$ and $\log(1/\Delta)$ factors. (The proof appears in the full version of the paper [1].)

**Theorem 8.** *The adaptive MW variant in Eq. (5) with $\eta_t = \sqrt{\log(N)/t}$ in the stochastic regime (with no corruption), achieves constant $O(\Delta^{-1} \log N \log^2(\Delta^{-1} \log N))$ expected pseudo regret for any $T$.*

On the other hand, we give a simple example which demonstrates that the OMD variant of MW exhibits a strictly inferior performance compared to the FTRL variant (see Eq. (4)) when adversarial corruptions are present. For simplicity, assume that the corruption level $C$ is a positive integer. Consider the following corrupted stochastic instance with $K = 2$ experts. The mean loss of expert #1 is $\mu_1 = \frac{1}{2}(1 - \Delta)$ while the mean loss of expert #2 is $\mu_1 = \frac{1}{2}(1 + \Delta)$. The adversary introduces corruption over the first $C$ rounds, and modifies the first $C$ losses of expert #1 to 1's and those of expert #2 to 0's.

For this simple problem instance, we show the following (see the full version of the paper [1] for the proof).

**Theorem 9.** *The expected pseudo regret of the adaptive MW algorithm in Eq. (5) with $\eta_t = \alpha/\sqrt{t}$ where $\alpha = \Omega(1/\sqrt{C})$ on the instance described above for $T \geq T_1 = \Theta\big(\min\{C/\Delta^2, \exp(\sqrt{C}/\alpha)\}\big)$ rounds is at least $\Omega(\Delta T_1)$.*

In particular, if the learner does not have non-trivial bounds on the corruption level $C$ and gap $\Delta$ (that is, $\alpha$ is a constant independent of $C$ and $\Delta$), then the regret is necessarily at least $\Omega(C/\Delta)$ or is exponentially large in $\sqrt{C}$.

## 4 Numerical Simulations

We conducted a basic numerical experiment to illustrate our regret bounds and the gap between OMD and FTRL discussed above. The experiment setup consists of two experts with different gaps $\Delta \in \{0.05, 0.15, 0.25, 0.4\}$. The losses were taken as Bernoullis and the corruption strategy injected contamination in the first rounds up to a total budget of $C$, inflicting maximal loss on the best expert while zeroing the losses of the other expert.

The results, shown in Fig. 1, demonstrate that for the stochastic case without corruption ($C = 0$) OMD achieves better pseudo regret, but is substantially outperformed by FTRL when $C > 0$. In Fig. 2 we further show the inverse dependence of the pseudo-regret on the minimal gap $\Delta$, which precisely supports our theoretical finding discussed in Section 3.3.

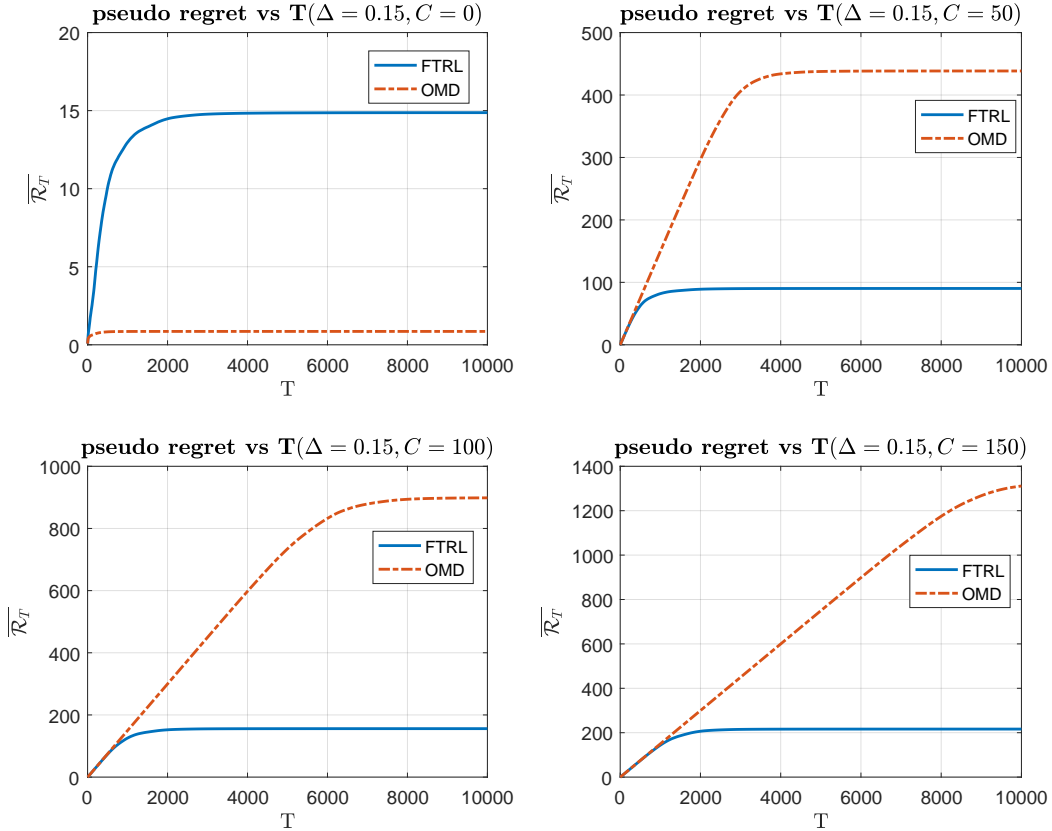

Figure 1: Pseudo regret of the two variants of MW, as a function of the number of rounds $T$ for different corruption levels $C$.

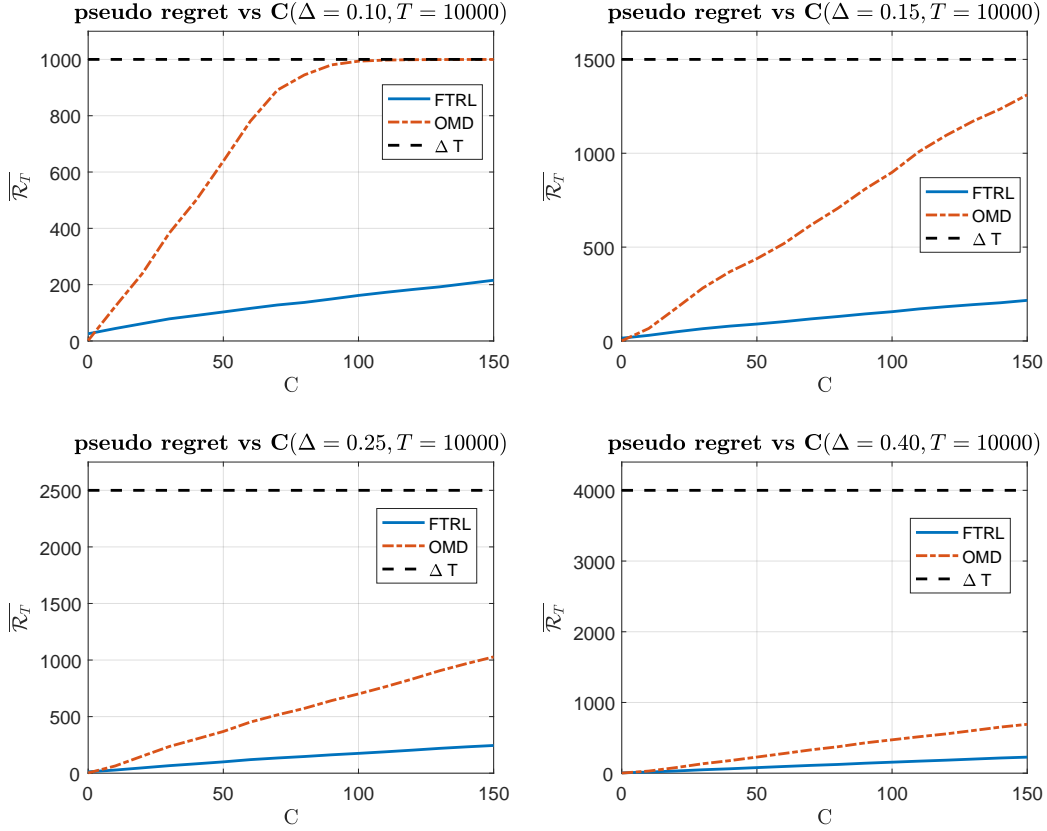

Figure 2: Pseudo regret of the two variants of MW, as a function of corruption level $C$ for different values of the gap $\Delta$.

## Broader Impact

There are no foreseen ethical or societal consequences for the research presented herein.

## Acknowledgments and Disclosure of Funding

We thank Alon Cohen for helpful discussions. This project has received funding from the European Research Council (ERC) under the European Union's Horizon 2020 research and innovation program (grant agreement No. 882396), from the Israel Science Foundation (grants 2549/19; 993/17; 2188/20), from the Yandex Initiative in Machine Learning and partially funded by an unrestricted gift from Google. Any opinions, findings, and conclusions or recommendations expressed in this work are those of the author(s) and do not necessarily reflect the views of Google.

## Footnotes

[2]More precisely, the algorithm derived from OMD achieves a near-optimal (yet still constant, independent of $T$) bound, which is tight up to $\log \log N$ and $\log(1/\Delta)$ factors.

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
