[Supplementary Material]

# A Proofs

## A.1 Preliminaries: Online optimization with time-dependent regularization

We give a brief background on Follow the Regularized Leader and Online Mirror Descent algorithmic templates, in the case where the regularization is varying and time-dependent.

The setup is the standard setup of online linear optimization. Let $\mathcal{W} \subseteq \mathbb{R}^d$ be a convex domain. On each prediction round $t = 1, \ldots, T$, the learner has to produce a prediction $w_t \in \mathbb{R}^d$ based on $g_1, \ldots, g_{t-1}$, and subsequently observes a new loss vector $g_t$ and incurs the loss $w_t \cdot g_t$. The goal is to minimize the regret compared to any $w^\star \in \mathcal{W}$, given by $\sum_{t=1}^{T} g_t \cdot (w_t - w^\star)$.

**Follow the Regularized Leader (FTRL).** The FTRL template generates predictions $w_1, \ldots, w_T \in \mathcal{W}$, for $t = 1, \ldots, T$, as follows:

$$w_t = \operatorname*{argmin}_{w \in \mathcal{W}} \left\{ w \cdot \sum_{s=1}^{t-1} g_s + R_t(w) \right\}. \tag{6}$$

Here, $R_1, \ldots, R_T : \mathcal{W} \to \mathbb{R}$ is a sequence of twice-differentiable, strictly convex functions.

The derivation and analysis of FTRL-type algorithms is standard; see, e.g., [29, 13, 25]. In our analysis, however, we require a particular regret bound that we could not find stated explicitly in the literature (similar bounds exist, however, and date back at least to [7]). For completeness, we provide the bound here with a proof in the full version of the paper [? ].

**Theorem 10.** *Suppose that $R_t = \eta_t^{-1} R$ for all $t$ for some strictly convex $R$, with $\eta_1 \geq \ldots \geq \eta_T > 0$. Then there exists a sequence of points $z_t \in [w_t, w_{t+1}]$ such that the following regret bound holds for all $w^\star \in \mathcal{W}$:*

$$\sum_{t=1}^{T} g_t \cdot (w_t - w^\star) \leq \frac{1}{\eta_1} \big( R(w^\star) - R(w_1) \big) + \sum_{t=1}^{T} \Big( \frac{1}{\eta_{t+1}} - \frac{1}{\eta_t} \Big) \big( R(w^\star) - R(w_{t+1}) \big) + \frac{1}{2} \sum_{t=1}^{T} \eta_t \big( \|g_t\|_t^* \big)^2,$$

*where $\|g\|_t^2 = g^\mathsf{T} \nabla^2 R(z_t) g$ is the local norm induced by $R$ at an appropriate $z_t \in [w_t, w_{t+1}]$, and $\| \cdot \|_t^*$ is its dual norm.*

**Online Mirror Descent (OMD).** The closely-related OMD framework produces predictions $w_1, \ldots, w_T$ via the following procedure: initialize $w_1 = \operatorname{argmin}_{w \in \mathcal{W}} R_1(w)$, and for $t = 1, \ldots, T$, compute

$$w'_{t+1} = \operatorname*{argmin}_{w} \big\{ g_t \cdot w + D_{R_t}(w, w_t) \big\} = (\nabla R_t)^{-1} \big( \nabla R_t(w_t) - g_t \big);$$

$$w_{t+1} = \operatorname*{argmin}_{w \in \mathcal{W}} D_{R_t}(w, w'_{t+1}). \tag{7}$$

Here, $R_1, \ldots, R_T : \mathcal{W} \to \mathbb{R}$ is a sequence of twice-differentiable, strictly convex functions and $D_R(w', w) = R(w') - R(w) - \nabla R(w) \cdot (w' - w)$ is the Bregman divergence of a convex function $R$ at point $w \in \mathcal{W}$.

The proof of the following regret bound (which is again a somewhat specialized variant of standard bounds for OMD) appears in the full version of the paper [? ].

**Theorem 11.** *Suppose that $R_t = \eta_t^{-1} R$ for all $t$ for some strictly convex $R$, with $\eta_1 \geq \ldots \geq \eta_T > 0$. Then there exists a sequence of points $z_t \in [w_t, w'_{t+1}]$ such that the following regret bound holds for all $w^\star \in \mathcal{W}$:*

$$\sum_{t=1}^{T} g_t \cdot (w_t - w^\star) \leq \frac{1}{\eta_1} \big( R(w^\star) - R(w_1) \big) + \sum_{t=1}^{T-1} \Big( \frac{1}{\eta_{t+1}} - \frac{1}{\eta_t} \Big) D_R(w^\star, w_{t+1}) + \frac{1}{2} \sum_{t=1}^{T} \eta_t \big( \|g_t\|_t^* \big)^2,$$

*where $\| \cdot \|_t$ is the local norm induced by $R$ at an appropriate $z_t \in [w_t, w'_{t+1}]$, and $\| \cdot \|_t^*$ is its dual.*

## A.2 Upper bounds for FTRL

**Proof of Lemma 6.** We observe that Eq. (4) is an instantiation of FTRL with $R_t(p) = \eta_t^{-1} R(p)$ as regularizations, where $R(p) = -H(p) = \sum_{i=1}^{N} p_i \log p_i$ is the negative entropy. Hence, we can invoke

Theorem 10 to bound the regret compared to any probability distribution $p^\star$. It suffices to bound the regret for $p^\star$ that minimizes $\sum_{t=1}^{T} p \cdot \ell_t$, which is always a point-mass on a single expert $i^\star$, for which $R(p^\star) = 0$. Therefore, Theorem 10 in our case reads

$$\sum_{t=1}^{T} \sum_{i=1}^{N} p_{t,i}(g_{t,i} - g_{t,i^\star}) \leq -\frac{1}{\eta_1} R(p_1) - \sum_{t=1}^{T} \left( \frac{1}{\eta_{t+1}} - \frac{1}{\eta_t} \right) R(p_{t+1}) + \frac{1}{2} \sum_{t=1}^{T} \eta_t \left( \|g_t\|_t^* \right)^2.$$

Now set $\eta_t = \sqrt{\log(N)/t}$. For the first two terms in the bound, observe that $R(p_1) = -\log N$, and further, that

$$\frac{1}{\eta_{t+1}} - \frac{1}{\eta_t} = \frac{1}{\sqrt{\log N}} \frac{1}{\sqrt{t} + \sqrt{t+1}} \leq \frac{1}{2\sqrt{t \log N}} = \frac{\eta_t}{2 \log N}. \tag{8}$$

For the final sum, we have to evaluate the Hessian $\nabla^2 R(p_t')$ at a point $p_t' \in [p_t, p_{t+1}]$. A straightforward differentiation shows that this matrix is diagonal, with diagonal elements $\nabla^2 R(p_t')_{ii} = 1/p_{t,i}'$. Thus,

$$\left( \|g_t\|_t^* \right)^2 = g_t^\top \left( \nabla^2 R(p_t') \right)^{-1} g_t = p_t' \cdot g_t^2. \tag{9}$$

The final sum can be divided and bounded as follows

$$\sum_{t=1}^{T} \eta_t \left( p_t' \cdot g_t^2 \right) = \sum_{t=1}^{4 \log N} \eta_t \left( p_t' \cdot g_t^2 \right) + \sum_{t=1+4 \log N}^{T} \eta_t \left( p_t' \cdot g_t^2 \right)$$

$$\leq 4 \log N + \sum_{t=1+\log N}^{T} \eta_t \left( p_t' \cdot g_t^2 \right).$$

Where we used the fact that $\sum_{s=1}^{t} \eta_s = \sum_{s=1}^{t} \sqrt{\log(N)/s} \leq 2\sqrt{t \log N}$. To conclude the proof it suffices to show that $p_{t,i}' \leq 9 p_{t,i}$ for $t \geq 4 \log N$. To see this, denote $G_t = \sum_{s=1}^{t-1} g_s$ and write

$$\frac{e^{-\eta_{t+1} G_{t+1,i}}}{e^{-\eta_t G_{t,i}}} = e^{-\eta_{t+1} g_{t,i}} e^{(\eta_t - \eta_{t+1}) G_{t,i}}.$$

For $t \geq 4 \log N$, the following relations hold:

$$0 < \eta_{t+1} |g_{t,i}| \leq \eta_{t+1} \leq \frac{1}{2};$$

$$0 < (\eta_t - \eta_{t+1}) |G_{t,i}| \leq \sqrt{\log N} \frac{\sqrt{t+1} - \sqrt{t}}{\sqrt{t(t+1)}} t \leq \frac{\sqrt{\log N}}{\sqrt{t} + \sqrt{t+1}} \leq \eta_t \leq \frac{1}{2}.$$

Hence, for $t \geq 4 \log N$ we have

$$\frac{1}{3} \leq \frac{e^{-\eta_{t+1} G_{t+1,i}}}{e^{-\eta_t G_{t,i}}} \leq 3,$$

and consequently

$$p_{t+1,i} = \frac{e^{-\eta_{t+1} G_{t+1,i}}}{\sum_{j=1}^{N} e^{-\eta_{t+1} G_{t+1,j}}} \leq 9 \frac{e^{-\eta_t G_{t,i}}}{\sum_{j=1}^{N} e^{-\eta_t G_{t,j}}} = 9 p_{t,i}.$$

Since $p_t' \in [p_t, p_{t+1}]$, the same inequality holds for $p_t'$; that is, $p_{t,i}' \leq 9 p_{t,i}$ for all $i$, and the proof is complete. ∎

**Lemma 12.** *For the adaptive MW algorithm in Eq. (4) with loss vectors $g_t = \tilde{\ell}_{t,i}$, we have*

$$\sum_{t=1}^{T} \eta_t \sum_{i=1}^{N} p_{t,i} (\tilde{\ell}_{t,i} - \tilde{\ell}_{t,i^\star})^2 \leq \frac{16 \log N}{\Delta} + \frac{1}{8} \overline{\mathcal{R}}_T.$$

**Proof.** By setting $t_0 = 64\Delta^{-2}\log N$ and $\eta_t = \sqrt{\log(N)/t}$ we obtain

$$\sum_{t=1}^{T}\eta_t\sum_{i=1}^{N}p_{t,i}\big(\tilde{\ell}_{t,i}-\tilde{\ell}_{t,i^\star}\big)^2 \le \sum_{t=1}^{t_0}\eta_t + \sum_{t=t_0+1}^{T}\eta_{t_0}\sum_{i=1}^{N}p_{t,i}\big(\tilde{\ell}_{t,i}-\tilde{\ell}_{t,i^\star}\big)^2$$

$$\le 2\sqrt{\log(N)}\sqrt{t_0} + \frac{\Delta}{8}\sum_{t=t_0+1}^{T}\sum_{i=1}^{N}p_{t,i}\big(\tilde{\ell}_{t,i}-\tilde{\ell}_{t,i^\star}\big)^2$$

$$\le \frac{16\log N}{\Delta} + \frac{1}{8}\sum_{t=t_0+1}^{T}\sum_{i=1}^{N}p_{t,i}\big(\mu_i-\mu_{i^\star}\big),$$

where in the final inequality we used observation 3. To conclude we note that $p_{t,i}(\mu_i - \mu_{i^\star}) \ge 0$, thus we can modify the last summation to range over $t = 1, \ldots, T$. ∎

**Lemma 13.** *For the adaptive MW algorithm in Eq.* (4), *we have*

$$\frac{1}{\log N}\sum_{t=1}^{T}\eta_t H(p_{t+1}) \le \frac{50\log N}{\Delta} + \frac{5}{8}\overline{\mathcal{R}}_T.$$

**Proof.** First we split the sum as follows,

$$\frac{1}{\log N}\sum_{t=1}^{T}\eta_t H(p_{t+1}) = \frac{1}{\log N}\sum_{t=1}^{t_0}\eta_t H(p_{t+1}) + \frac{1}{\log N}\sum_{t=t_0+1}^{T}\eta_t H(p_{t+1}),$$

where $t_0 = 64\Delta^{-2}\log N$. For the summation of $t = \{t_0 + 1, \ldots, T\}$ we use Lemma 7 with $\tau = t\log N \ge t_0\log N = 64\Delta^{-2}\log^2 N$ to obtain

$$\frac{1}{\log N}\sum_{t=t_0+1}^{T}\eta_t H(p_{t+1}) = \sum_{t=t_0+1}^{T}\frac{1}{\sqrt{t\log N}}\sum_{i=1}^{N}p_{t+1,i}\log\frac{1}{p_{t+1,i}}$$

$$\le \frac{5}{8}\sum_{t=t_0+1}^{T}\sum_{i\neq i^\star}p_{t+1,i}\Delta + 2\sum_{t=t_0+1}^{T}\frac{1}{\sqrt{t\log N}}e^{-\frac{1}{8}\Delta\sqrt{t\log N}}$$

$$\le \frac{5}{8}\sum_{t=t_0+1}^{T}\sum_{i=1}^{N}p_{t,i}\big(\mu_i-\mu_{i^\star}\big) + \Delta + 2\sum_{t=t_0+1}^{T}\frac{1}{\sqrt{t\log N}}e^{-\frac{1}{8}\Delta\sqrt{t\log N}},$$

where the last inequality follows for reordering terms in the summation and that $\Delta \le \mu_i - \mu_{i^\star}$ for $i \neq i^\star$. Using the fact that $p_{t,i}(\mu_i - \mu_{i^\star}) \ge 0$ we get

$$\frac{1}{\log N}\sum_{t=t_0+1}^{T}\eta_t H(p_{t+1}) \le \frac{5}{8}\sum_{t=1}^{T}\sum_{i=1}^{N}p_{t,i}\big(\mu_i-\mu_{i^\star}\big) + \Delta + 2\sum_{t=t_0+1}^{T}\frac{1}{\sqrt{t\log N}}e^{-\frac{1}{8}\Delta\sqrt{t\log N}}. \tag{10}$$

Moreover, we have

$$\sum_{t=t_0+1}^{T}\frac{1}{\sqrt{t\log N}}e^{-\frac{1}{8}\Delta\sqrt{\log N}\sqrt{t}} \le \frac{1}{\sqrt{\log N}}\int_{t_0}^{T}\frac{1}{\sqrt{t}}e^{-\frac{1}{8}\Delta\sqrt{\log N}\sqrt{t}}dt$$

$$= \frac{1}{\sqrt{\log N}}\cdot\frac{16}{\Delta\sqrt{\log N}}e^{-\frac{1}{8}\Delta\sqrt{\log N}\sqrt{t}}\Big|_{T}^{t_0} \tag{11}$$

$$\le \frac{16}{\Delta\log N}$$

$$\le \frac{16}{\Delta}.$$

Lastly, for the summation of $t = \{1, \ldots, t_0\}$ we get

$$\frac{1}{\log N}\sum_{t=1}^{t_0}\eta_t H(p_{t+1}) \le 2\sqrt{t_0\log N} = \frac{16\log N}{\Delta} \tag{12}$$

which follows from $H(p) \le \log N$ and $\sum_{t=1}^{t_0}1/\sqrt{t} \le 2\sqrt{t_0}$. Combining Eqs. (10) to (12), the proof is concluded. ∎

## A.3 Lower bound for OMD

**Proof of Theorem 9.** Let $q_t$ denote the probability that MW-OMD chooses the best expert (i.e., expert #1) on round $t$. For $t \leq C$, the best expert suffers higher losses than the other expert, thus $\mathbf{E}[q_t] \leq 1/2$. For $t > C$, it holds that

$$q_t = \frac{e^{-\sum_{s=1}^{t-1} \eta_s(\tilde{\ell}_{s,1} - \tilde{\ell}_{s,2})}}{1 + e^{-\sum_{s=1}^{t-1} \eta_s(\tilde{\ell}_{s,1} - \tilde{\ell}_{s,2})}} \leq e^{-\sum_{s=1}^{t-1} \eta_s(\tilde{\ell}_{s,1} - \tilde{\ell}_{s,2})} = e^{-\sum_{s=1}^{C} \eta_s} \exp\left( \sum_{s=C+1}^{t-1} \eta_s(\ell_{s,2} - \ell_{s,1}) \right).$$

Now, observe that

$$\sum_{s=1}^{C} \eta_s \geq C \eta_C = \alpha\sqrt{C}.$$

Also, by a standard application of Hoeffding's lemma (e.g., Appendix A of [3]),

$$\mathbf{E}\exp\left( \sum_{s=C+1}^{t-1} \eta_s(\ell_{s,2} - \ell_{s,1}) \right) = \prod_{s=C+1}^{t-1} \mathbf{E}e^{\eta_s(\ell_{s,2} - \ell_{s,1})}$$

$$\leq \prod_{s=C+1}^{t-1} e^{\eta_s \Delta + \eta_s^2/8}$$

$$\leq \exp\left( \Delta \sum_{s=1}^{t-1} \eta_s \right) \exp\left( \frac{1}{8} \sum_{s=1}^{t-1} \eta_s^2 \right)$$

$$\leq \exp\left( 2\alpha\Delta\sqrt{t} + \alpha^2 \log t \right).$$

Overall, we have shown that for $t > C$,

$$\mathbf{E}[q_t] \leq \exp\left( -\alpha\sqrt{C} + 2\alpha\Delta\sqrt{t} + \alpha^2 \log t \right).$$

Whenever $t \leq t_1 := \min\left\{ 2^{-6}C/\Delta^2, \exp(\frac{1}{4}\sqrt{C}/\alpha) \right\}$, the right hand side is $\leq \exp(-\frac{1}{2}\alpha\sqrt{C}) \leq \frac{1}{2}$ for $\alpha \geq 1/\sqrt{C}$. Hence, in that case,

$$\mathcal{R}_T \geq \sum_{s=1}^{t_1} \Delta\mathbf{E}[1 - q_s] \geq \sum_{s=1}^{t_1} \frac{1}{2}\Delta \geq \frac{1}{2}\Delta t_1. \qquad \blacksquare$$

# B  Analysis of OMD in the Purely Stochastic Case

**Proof of Theorem 8.** Applying Theorem 11 for the experts setting we get

$$\mathcal{R}_T \leq \frac{1}{\eta_1}\left( H(p_1) - H(p^\star) \right) + \sum_{t=1}^{T-1}\left( \frac{1}{\eta_{t+1}} - \frac{1}{\eta_t} \right) \sum_{i=1}^{N} p_i^\star \log \frac{p_i^\star}{p_{t+1,i}} + \frac{1}{2}\sum_{t=1}^{T} \eta_t \left( \|\ell_t\|_t^* \right)^2,$$

where we used the fact that the Bregman divergence of the negative entropy is the KL divergence. In addition, using similar observations as in the proof of Lemma 6 (e.g., Eqs. (8) and (9)) and setting $\eta_t = c/\sqrt{t}$ we obtain

$$\mathcal{R}_T \leq \frac{\log N}{c} + \frac{1}{2c^2}\sum_{t=1}^{T-1} \eta_t \log \frac{1}{p_{t+1,i^\star}} + \frac{1}{2}\sum_{t=1}^{T}\sum_{i=1}^{N} \eta_t p_{t,i}\ell_{t,i}^2.$$

Applying additive translation we get,

$$\mathcal{R}_T \leq \frac{\log N}{c} + \frac{1}{2c^2}\sum_{t=1}^{T-1} \eta_t \log \frac{1}{p_{t+1,i^\star}} + \frac{1}{2}\sum_{t=1}^{T} \eta_t \sum_{i=1}^{N} p_{t,i}(\ell_{t,i} - \ell_{t,i^\star})^2. \qquad (13)$$

Similarly to Lemma 12 we can bound the third term by

$$\frac{1}{2}\sum_{t=1}^{T} \eta_t \sum_{i=1}^{N} p_{t,i}(\ell_{t,i} - \ell_{t,i^\star})^2 \leq \frac{c^2}{\Delta} + \frac{1}{2}\overline{\mathcal{R}}_T = \frac{\log N}{\Delta} + \frac{1}{2}\overline{\mathcal{R}}_T. \qquad (14)$$

We now examine the second term. Using the MW algorithm defined in Eq. (5) we have,

$$\log \frac{1}{p_{t+1,i^\star}} = \log \frac{\sum_{i=1}^{N} e^{-\sum_{s=1}^{t-1} \eta_s \ell_{t,i}}}{e^{-\sum_{s=1}^{t-1} \eta_s \ell_{s,i^\star}}} = \log\left(1 + \sum_{i \neq i^\star} e^{-\sum_{s=1}^{t-1} \eta_s (\ell_{s,i} - \ell_{s,i^\star})}\right).$$

Plugging it back to the original term we get

$$\frac{1}{2c^2} \sum_{t=1}^{T-1} \eta_t \log \frac{1}{p_{t+1,i^\star}} = \frac{1}{2c} \sum_{t=1}^{T-1} \frac{1}{\sqrt{t}} \log\left(1 + \sum_{i \neq i^\star} e^{-\sum_{s=1}^{t-1} \eta_s (\ell_{s,i} - \ell_{s,i^\star})}\right).$$

By taking the expectation and using its linearity property we obtain

$$\frac{1}{2c} \sum_{t=1}^{T-1} \frac{1}{\sqrt{t}} \mathbf{E}\left[\log\left(1 + \sum_{i \neq i^\star} e^{-\sum_{s=1}^{t-1} \eta_s (\ell_{t,i} - \ell_{t,i^\star})}\right)\right] \leq \frac{1}{2c} \sum_{t=1}^{T-1} \frac{1}{\sqrt{t}} \log\left(1 + \sum_{i \neq i^\star} \mathbf{E}\left[e^{-\sum_{s=1}^{t-1} \eta_s (\ell_{s,i} - \ell_{s,i^\star})}\right]\right)$$

$$\leq \frac{1}{2c} \sum_{t=1}^{T-1} \frac{1}{\sqrt{t}} \log\left(1 + \sum_{i \neq i^\star} \prod_{s=1}^{t-1} \mathbf{E}\left[e^{-\eta_s (\ell_{s,i} - \ell_{s,i^\star})}\right]\right),$$

where we used Jensen inequality for concave functions for the first inequality and the fact that $x_t := \ell_{t,i} - \ell_{t,i^\star}$ are i.i.d. for the second inequality. Applying Hoeffding's Lemma yields,

$$\frac{1}{2c} \sum_{t=1}^{T-1} \frac{1}{\sqrt{t}} \log\left(1 + \sum_{i \neq i^\star} \prod_{s=1}^{t-1} \mathbf{E}\left[e^{-\eta_s (\ell_{s,i} - \ell_{s,i^\star})}\right]\right) \leq \frac{1}{2c} \sum_{t=1}^{T-1} \frac{1}{\sqrt{t}} \log\left(1 + N \exp\left(\sum_{s=1}^{t-1} \left(\tfrac{1}{2}\eta_s^2 - \eta_s \Delta\right)\right)\right).$$

Next, we bound the argument of the exponent

$$\sum_{s=1}^{t-1} \left(\tfrac{1}{2}\eta_s^2 - \eta_s \Delta\right) \leq \frac{c^2}{2} \sum_{s=1}^{t-1} \frac{1}{s} - c\Delta \sum_{s=1}^{t-1} \frac{1}{\sqrt{s}}$$

$$\leq \frac{c^2}{2}(1 + \log t) - c\Delta\sqrt{t}$$

$$\leq c^2 \log t - c\Delta\sqrt{t},$$

where we bounded the summations by their integrals. Therefore we have

$$\frac{1}{2c} \sum_{t=1}^{T-1} \frac{1}{\sqrt{t}} \log\left(1 + N e^{\sum_{s=1}^{t-1}\left(\frac{\eta_s^2}{2} - \eta_s \Delta\right)}\right) \leq \frac{1}{2c} \sum_{t=1}^{T-1} \frac{1}{\sqrt{t}} \log\left(1 + N e^{c^2 \log t - c\Delta\sqrt{t}}\right).$$

First we examine the sum from $t_1$ onward, while we require that for $t \geq t_1$ it holds

$$c^2 \log t \leq \tfrac{1}{2}c\Delta\sqrt{t}. \tag{15}$$

To satisfy Eq. (15) it suffices to take

$$t_1 = \left(\frac{8c}{\Delta}\right)^2 \log^2 \frac{8c}{\Delta}.$$

Therefore,

$$\frac{1}{2c} \sum_{t=t_1+1}^{T-1} \frac{1}{\sqrt{t}} \log\left(1 + N e^{c^2 \log t - 2c\Delta\sqrt{t}}\right) \leq \frac{1}{2c} \sum_{t=t_1+1}^{T-1} \frac{1}{\sqrt{t}} \log\left(1 + N e^{-\frac{1}{2}c\Delta\sqrt{t}}\right)$$

$$\leq \frac{N}{2c} \sum_{t=t_1+1}^{T-1} \frac{1}{\sqrt{t}} e^{-\frac{1}{2}c\Delta\sqrt{t}} \qquad (\log(1+x) \leq x)$$

$$\leq \frac{N}{2c} \int_{t_1}^{T-1} \frac{1}{\sqrt{t}} e^{-\frac{1}{2}c\Delta\sqrt{t}} dt$$

$$\leq \frac{N}{c^2 \Delta} e^{-\frac{1}{2}c\Delta\sqrt{t_1}} dt$$

$$\leq \frac{2N}{c^2 \Delta} e^{-c^2 \log t_1} \qquad (c^2 \log t_1 \leq \tfrac{1}{2}c\Delta\sqrt{t_1})$$

$$\leq \frac{2}{\Delta \log N}. \qquad (t_1 > 1 \text{ and } c = \sqrt{\log N})$$

To conclude we examine the bound up to $t_1$,

$$\frac{1}{2c}\sum_{t=1}^{t_1}\frac{1}{\sqrt{t}}\log\Big(1+Ne^{c^2\log t-2c\Delta\sqrt{t}}\Big) \le \frac{1}{2c}\sum_{t=1}^{t_1}\frac{1}{\sqrt{t}}\log\Big(2Ne^{c^2\log t-2c\Delta\sqrt{t}}\Big)$$

$$\le \frac{1}{2c}\sum_{t=1}^{t_1}\frac{1}{\sqrt{t}}\Big(\log 2N + c^2\log t\Big)$$

$$\le \frac{\log 2N + c^2\log t_1}{2c}\sum_{t=1}^{t_1}\frac{1}{\sqrt{t}} \qquad (\log t \le \log t_1)$$

$$\le \frac{\log 2N + c^2\log t_1}{c}\sqrt{t_1}.$$

Since $c = \sqrt{\log N}$, for $t_1 \ge \lceil e^2 \rceil$ we have $c^2\log t_1 = \log N\log t_1 \ge \log 2N$, and also Eq. (15) still holds. This implies

$$\frac{\log 2N + c^2\log t_1}{c}\sqrt{t_1} \le 2c\log t_1\sqrt{t_1}$$

$$\le 2\Delta t_1$$

$$\le 128\frac{\log N}{\Delta}\log^2\Big(\frac{8\sqrt{\log N}}{\Delta}\Big).$$

when we used the fact that $c\log t_1 \le \Delta\sqrt{t_1}$ for the last inequality. Adding both results(up to $t_1$ and from $t_1$ onward) we obtain,

$$\frac{1}{2c^2}\sum_{t=1}^{T-1}\eta_t\log\frac{1}{p_{t+1,i^\star}} \le 128\frac{\log N}{\Delta}\log^2\Big(\frac{8\sqrt{\log N}}{\Delta}\Big) + \frac{2}{\Delta\log N} \qquad (16)$$

Finally, plugging Eqs. (14) and (16) into Eq. (13), taking the expectation and rearranging terms we get

$$\mathbf{E}\big[\overline{\mathcal{R}}_T\big] \le \frac{256\log N}{\Delta}\log^2\Big(\frac{8\log N}{\Delta}\Big) + \frac{8\log N}{\Delta}. \qquad \blacksquare$$

## C   Analysis of Time-varying Regularization Algorithms

In this section, we assume the setup of online (linear) optimization, with the notation established in Section 4.1. For the proofs below, we recall the notion of a Bregman divergence. For a continuously differentiable and strictly convex function $F : \mathcal{W} \to \mathbb{R}$ defined on a closed convex set $\mathcal{W}$, the Bregman divergence associated with $F$ at a point $w \in \mathcal{W}$ is defined by

$$\forall w' \in \mathcal{W}, \qquad D_F(w', w) = F(w') - F(w) - \nabla F(w)\cdot(w' - w).$$

### C.1   Follow the Regularized Leader

First, we present a general analysis for Follow the Regularized Leader, described in Eq. (6), and later establish Theorem 10.

**Theorem 14.** *There exists a sequence of points $z_t \in [w_t, w_{t+1}]$ such that, for all $w^\star \in \mathcal{W}$,*

$$\sum_{t=1}^{T} g_t\cdot(w_t - w^\star) \le R_{T+1}(w^\star) - R_1(w_1) + \sum_{t=1}^{T}\big(R_t(w_{t+1}) - R_{t+1}(w_{t+1})\big) + \frac{1}{2}\sum_{t=1}^{T}\big(\|g_t\|_t^*\big)^2.$$

*Here $\|w\|_t = \sqrt{w^\mathsf{T}\nabla^2 R_t(z_t)w}$ is the local norm induced by $R_t$ at $z_t$, and $\|\cdot\|_t^*$ is its dual.*

**Proof.** Denote $\Phi_t(w) = w\cdot\sum_{s=1}^{t-1} g_s + R_t(w)$, so that $w_t = \operatorname{argmin}_{w\in\mathcal{W}}\Phi_t(w)$. We first write

$$\sum_{t=1}^{T} g_t\cdot w_{t+1} = \sum_{t=1}^{T}\big(\Phi_{t+1}(w_{t+1}) - \Phi_t(w_{t+1})\big) + \sum_{t=1}^{T}\big(R_t(w_{t+1}) - R_{t+1}(w_{t+1})\big)$$

$$= \Phi_{T+1}(w_{T+1}) - \Phi_1(w_1) + \sum_{t=1}^{T}\big(\Phi_t(w_t) - \Phi_t(w_{t+1})\big) + \sum_{t=1}^{T}\big(R_t(w_{t+1}) - R_{t+1}(w_{t+1})\big).$$

Since $w_t$ is the minimizer of $\Phi_t$ over $\mathcal{W}$, first-order optimality conditions imply

$$\Phi_t(w_t) - \Phi_t(w_{t+1}) = -\nabla\Phi_t(w_t) \cdot (w_{t+1} - w_t) - D_{\Phi_t}(w_{t+1}, w_t) \leq -D_{\Phi_t}(w_{t+1}, w_t) = -D_{R_t}(w_{t+1}, w_t),$$

where we have used the fact that the Bregman divergence is invariant to linear terms. On the other hand, since $w_{T+1}$ is the minimizer of $\Phi_{T+1}$, we have that

$$\sum_{t=1}^{T} g_t \cdot w^{\star} = \Phi_{T+1}(w^{\star}) - R_{T+1}(w^{\star}) \geq \Phi_{T+1}(w_{T+1}) - R_{T+1}(w^{\star}).$$

Combining inequalities and observing that $\Phi_1(w_1) = R_1(w_1)$, we obtain

$$\sum_{t=1}^{T} g_t \cdot (w_{t+1} - w^{\star}) \leq R_{T+1}(w^{\star}) - R_1(w_1) + \sum_{t=1}^{T}\left(R_t(w_{t+1}) - R_{t+1}(w_{t+1})\right) - \sum_{t=1}^{T} D_{R_t}(w_{t+1}, w_t).$$

On the other hand, a Taylor expansion of $R_t(\cdot)$ around $w_t$ with an explicit second-order remainder term implies that, for some intermediate point $z_t \in [w_t, w_{t+1}]$, it holds that

$$D_{R_t}(w_{t+1}, w_t) = \tfrac{1}{2}(w_{t+1} - w_t)^{\mathsf{T}} \nabla^2 R_t(z_t) (w_{t+1} - w_t) = \tfrac{1}{2}\|w_{t+1} - w_t\|_t^2.$$

An application of Holder's inequality then gives

$$g_t \cdot (w_t - w_{t+1}) \leq \|g_t\|_t^* \|w_t - w_{t+1}\|_t \leq \tfrac{1}{2}\left(\|g_t\|_t^*\right)^2 + \tfrac{1}{2}\|w_t - w_{t+1}\|_t^2 = \tfrac{1}{2}\left(\|g_t\|_t^*\right)^2 + D_{R_t}(w_{t+1}, w_t).$$

The proof is finalized by summing over $t = 1, \ldots, T$ and adding to the inequality above. $\blacksquare$

**Proof of Theorem 10.** Fix any $w^{\star} \in \mathcal{W}$. Observe that FTRL with regularizations $R_t(w) = \eta_t^{-1} R(w)$ is equivalent to FTRL with $R_t(w) = \eta_t^{-1}(R(w) - R(w^{\star}))$. Applying Theorem 14 for the latter and rearranging, we obtain the claimed bound. $\blacksquare$

## C.2 Online Mirror Descent

We next consider Online Mirror Descent (see Eq. (7)), and prove the following general bound from which Theorem 11 directly follows.

**Lemma 15.** *There exist points $z_t \in [w_t, w'_{t+1}]$ such that for all $w^{\star} \in \mathcal{W}$,*

$$\sum_{t=1}^{T} g_t \cdot (w_t - w^{\star}) \leq R_1(w^{\star}) - R_1(w_1) + \sum_{t=1}^{T-1}\left(D_{R_{t+1}}(w^{\star}, w_{t+1}) - D_{R_t}(w^{\star}, w_{t+1})\right) + \frac{1}{2}\sum_{t=1}^{T}\left(\|g_t\|_t^*\right)^2.$$

*Here $\|w\|_t = \sqrt{w^{\mathsf{T}} \nabla^2 R_t(z_t) w}$ is the local norm induced by $R_t$ at $z_t$, and $\|\cdot\|_t^*$ is its dual.*

**Proof.** Fix any $w^{\star} \in \mathcal{W}$. We will bound each of the terms $g_t \cdot (w_t - w^{\star})$. First, from the update rule of Mirror Descent and the three-point property of the Bregman divergence, we have

$$g_t \cdot (w'_{t+1} - w^{\star}) = (\nabla R(w_t) - \nabla R(w'_{t+1})) \cdot (w'_{t+1} - w^{\star})$$
$$= D_{R_t}(w^{\star}, w_t) - D_{R_t}(w^{\star}, w'_{t+1}) - D_{R_t}(w'_{t+1}, w_t).$$

Now, a Taylor expansion of $R_t$ at $x_t$ (with an explicit Lagrange remainder term) shows that there exists $z_t \in [w_t, w_{t+1}]$ for which

$$D_{R_t}(w'_{t+1}, w_t) = \tfrac{1}{2}(w'_{t+1} - w_t)^{\mathsf{T}} \nabla^2 R_t(z_t) (w'_{t+1} - w_t) = \tfrac{1}{2}\|w'_{t+1} - w_t\|_t^2.$$

Also, since $w_{t+1}$ is the projection (with respect to the Bregman divergence $R_t$) of the point $w'_{t+1}$ onto the set $\mathcal{W}$ that contains $w^{\star}$, it holds that $D_{R_t}(w^{\star}, w_{t+1}) \leq D_{R_t}(x^{\star}, w'_{t+1})$. Putting things together, we obtain

$$g_t \cdot (w'_{t+1} - w^{\star}) \leq D_{R_t}(w^{\star}, w_t) - D_{R_t}(w^{\star}, w_{t+1}) - \tfrac{1}{2}\|w'_{t+1} - w_t\|_{z_t}^2. \tag{17}$$

On the other hand, Hölder's inequality and the fact that $ab \leq \tfrac{1}{2}(a^2 + b^2)$ yield

$$g_t \cdot (w_t - w'_{t+1}) \leq \|g_t\|_t^* \cdot \|w_t - w'_{t+1}\|_t \leq \tfrac{1}{2}(\|g_t\|_t^*)^2 + \tfrac{1}{2}\|w_t - w'_{t+1}\|_t^2. \tag{18}$$

Summing Eqs. (17) and (18) together over $t = 1, \ldots, T$ gives the regret bound

$$\sum_{t=1}^{T} g_t \cdot (w_t - w^{\star}) \leq \sum_{t=1}^{T}\left(D_{R_t}(w^{\star}, w_t) - D_{R_t}(w^{\star}, w_{t+1})\right) + \frac{1}{2}\sum_{t=1}^{T}(\|g_t\|_t^*)^2.$$

Rearranging the first summation and using the facts that $D_{R_T}(w^{\star}, w_{T+1}) \geq 0$ and $D_{R_1}(w^{\star}, w_1) \leq R_1(w^{\star}) - R_1(w_1)$ (the latter follows since $w_1$ is the minimizer of $R_1$, and so $\nabla R_1(w_1) \cdot (w^{\star} - w_1) \geq 0$) gives the stated regret bound. $\blacksquare$