[Reviews · NeurIPS 2020]

Review 1

Summary and Contributions: ==post-rebuttal update== I am satisfied with the author's responses in their rebuttal and my score remains the same. In particular, regarding motivation for their own setting, I think the author’s response about malicious users affecting the feedback of a learning algorithm is great. Regarding the simulations, the author's explained why some of the plots were weird. Rather than taking out these plots, I think it's great to leave them in and just explain the effect (as the author's did in their rebuttal). =================== This paper studies decision-theoretic online learning (the Hedge setting), a full-information online learning game, under the assumption that the data is stochastic (with a gap in the mean between the best and second best experts) but only observed under adversarial corruption (with a parameter measuring the amount of corruption). To be clear, regret is defined based on the non-corrupted losses; it is only the observations that are potentially corrupted. The authors show that decreasing Hedge (the algorithm Hedge with the standard time-varying learning rate) obtains optimal regret (ignoring universal constants); as a warm-up step, they actually first showed that Hedge with an oracle tuning of the learning rate (by using information about the gap) also achieves optimal regret. Another result shows that, in the stochastic case with no corruption, OMD with time-varying learning rate (which subtly differs from Hedge with time-variying learning rate) obtains close to the optimal regret (but there is an extra log^2 term). Interestingly, though intuitively possible for those familiar with the downside of OMD in the adversarial setting, the authors show that coruption can force this algorithm to have very bad regret. Some experimental results are shown to support the theory.

Strengths: The theoretical results appear to be solid. I checked the proofs of Theorems 2 and 5 (the former is quite simple). I found the techniques used in the proof of Theorem 5 to be interesting. I do believe that this work is significant, because there should be important applications to this setting. The authors did not discuss any, but I'm sure they are there. One additional strength of this work is its exploration of OMD. I haven't previously seen attempts at analyzing OMD in the stochastic setting; the regret upper bound the authors obtain is not as the optimal one (which is achieved by both Follow-the-Leader (FTL) and decreasing Hedge). It would be interesting to see OMD is truly suboptimal in the non-corrupted stochastic-with-a-gap case, by showing a lower bound. The exploration of the upper bound in this work therefore will hopefully motivate others to find a lower bound in the future. The lower bound for OMD also is interesting, as are the empirical evaluations supporting the theory. I have some negative things to say about some of the plots in the weaknesses section.

Weaknesses: There are three main weaknesses of this work. All can be fixed for a camera-ready version (though the authors can subjectively decide whether or not to mention FTL). The first one is easy enough to fix; it is about (perceived) significance. I think the authors should have tried to motivate the setting more. It is more natural to have stochastic data that is corrupted where the regret is also based on the corrupted losses. The setting the authors consider, where the pseudo-regret (based on the non-corrupted stochastic model) is used, can still be motivated, but the authors didn't provide any motivation at all. Maybe this appears in previous works in the corruption literature, but still, something should be said in this paper. You definitely have the space for it (Observation 4's proof is extremely simple and I really do suggest moving it to the supplementary material). The second one is that the authors did not explore what would perhaps be the most natural algorithm for this setting: FTL. At least having some comments about whether FTL should be able to do well in this setting would be worthwhile, since we know that it obtains the optimal pseudo-regret (and I believe also expected regret) in the non-corrupted stochastic setting. Or, if FTL breaks, that would be interesting as well. The third one is about the presentation of the second set of plots (Figure 2). Something seems seriously wrong here. The titles and x-axis labels do not seem correct. While the results seem fine for OMD< we can see that for FTRL, it is actually doing *worse* as the gap increases. Please closely inspect this and mention in your rebuttal what is going on here.

Correctness: I carefully checked Theorems 2 and Theorem 5 (including all supporting results) and they are correct, modulo very minor corrections that I now mention: (i) Proof of Lemma 6 - In the supplementary material at the math display between lines 376 and 377, I believe that (1/3) and 3 should be replaced by e^{-2} and e^2 respectively. This is because on the RHS of the math display between lines 374 and 375, you do not know the sign of g_{t,i} and G_{t,i} and so must assume the worst case. This will affect some of the later constants (replace 9 by e^4 \leq 55). (ii) A brief explanation for how you handled the integral in equation (11) would be helpful. (iii) At the end of the proof of Lemma 7, on line 216, there is a small typo: replace "+ log N" by "- log N". Everything else is still correct though. For lack of time, I did not look at the proofs of Theorems 8 and 9. For Theorem 9, I'm familiar with this style of argument to break OMD and the result seems plausible.

Clarity: The paper is very clear.

Relation to Prior Work: The authors do well in relating to previous work, which here mostly involves works related to corruption in online learning. They seem to be well aware of all of the relevant literature.

Reproducibility: Yes

Additional Feedback: I don't have anything else to say that wasn't covered above.


Review 2

Summary and Contributions: AFTER REBUTTAL: The author response solidly addressed the concerns raised and I still advocate for the paper's acceptance. ============== The paper studies prediction with expert advice when the losses of each expert can be corrupted by an adversary who has a total corruption budget of C in infinity-norm and assuming that losses are in $[0,1]$ (this budget is unknown to the learner). This is motivated by two lines of work: one showing that multiplicative weights seamlessly provides gap-dependent guarantees when the two best experts are well separated; the other showing regret guarantees that are robust to adversarial corruptions in bandit learning. The main result is that follow-the-regularized leader with an adaptively tuned learning rate achieves the optimal regret of $O(log(N)/\Delta+C)$ -- Theorem 5. The first term is optimal even at the absence of corruptions while the second is necessary when the corruption affects only the feedback and not the loss incurred. Subsequently, the paper shows a separation between this algorithm and online mirror descent with an adaptive learning rate. Although in the uncorrupted setting both algorithms demonstrate the same performance guarantees (Theorem 8), at the existence of corruptions the regret of the latter algorithm is at least $C/Delta$ (Theorem 9). Intuitively, the reason behind this difference is that later rounds have discounted importance in online mirror descent while they are weighted in the same way with prior rounds in follow-the-regularized leader. In Section 4, the paper shows empirically that, although online mirror descent is better when C=0, its performance degrades heavily at the existence of corruptions; the figures support the distinction established in the theoretical results.

Strengths: The paper makes a number of interesting contributions to online learning. First, it shows a clean analysis that is robust to the existence of corruptions in the feedback. The analysis is very nicely described starting from a warm-up of known gap and moving then to the more intricate case of adaptive learning rate. Second, the lower bound on online mirror descent sheds additional light on the difference in the performance of the two algorithms (which are largely considered as equivalent). The experiments nicely illustrate this point. Overall, I think that this makes a nice contribution to the literature and I would be happy to see it accepted.

Weaknesses: I generally like the paper so I do not have significant concerns. One point that may merit further discussion is what happens when the corruption does not only affect the feedback but also the losses. By the classical multiplicative weights analysis, I assume that the result will never become worse than $\sqrt{T \log(N(}$. Is the linear dependence on $C$ necessary for $C<\sqrt{T \log(N)$?

Correctness: To the best of my understanding, the paper is technically sound.

Clarity: The paper is very well written. The paper positions the results nicely with respect to the previous literature.

Relation to Prior Work: The paper positions the results nicely with respect to the previous literature.

Reproducibility: Yes

Additional Feedback: One nit-picky comment: In page 13 (line 387), why do you get the additive $+\Delta$ term in the third line? Also is there any difference between Eq. (10) and the line above. Where do you use the fact that $p_{t,i}(\mu_i-\mu_{i^*}\geq 0$?


Review 3

Summary and Contributions: This is a theoretical paper that makes contributions to the area of protections with expert advice. The main contribution of the work is to show that multiplicative weights yields an optimal regret bound in the adversarially-corrupted stochastic setting. Interestingly, the paper also shows that online mirror descent does not match the optimal regret bound when corruptions are considered.

Strengths: The paper is relevant to NeurIPS and likely of broad interest to the online learning community. The main result makes a clear and crisp contribution to the field, showing that multiplicative weights obtains the optimal bound in the stochastic, corrupted, and adversarial regimes. The distinction between online mirror descent and follow the regularized leader is perhaps surprising, although suggested by results in [26].

Weaknesses: The contribution of the work seems to be primarily theoretical. While the corrupted regime is of theoretical interest, I am unclear about the practical impact of the result. It adds to our theoretical understanding of multiplicative weights but does not seem to yield a new insight for algorithm design or practical application. The proofs are careful and correct in my read, but seem to rely on established techniques developed in the stochastic case [32,33]. Thus, the technical contribution is limited.

Correctness: I read and verified the proofs provided.

Clarity: The paper is well-written and has a clear organization.

Relation to Prior Work: The relationship to prior work is clearly and correctly stated. The one area where additional context could be provided is in terms of relating the corruption model to other versions of "beyond worst-case" analysis in the online algorithms literature. There are other models where structure is placed on the type of adjustments an adversary can make to the input. A broad perspective and pointers beyond learning can be found at http://timroughgarden.org/w17/w17.html

Reproducibility: Yes

Additional Feedback: Thank you for your author response. I agree with your comments about the distinctions in your proof compared to prior work.


Review 4

Summary and Contributions: The paper develops algorithms to make predictions from corrupted expert advice, in a sequential (online) manner. While the losses are generated stochastically, and full loss vector is revealed to learner (full feedback problem), the losses are corrupted by an adversary, with maximum corruption budget parameterized by C. The adaptive multiplicative weights algorithm is shown to achieve optimal regret (up to constant factors). Authors also show online mirror descent algorithm is strictly inferior to the FTRL algorithm in this regime, though they are known to be strictly equivalent in non-corrupted loss regime. Post rebuttal: I have read the author feedback. I have kept my scores same.

Strengths: I would consider the following to be strengths of the paper: a. The paper is well written and technically strong. In fact, it is one of the rare theoretical paper which I could follow very comfortably, even though it is a technically heavy paper. The work does not introduce complex algorithms; instead it shows how the famous MW algorithm can be used in the corrupted expert advice regime. b. The work solves an interesting problem and does not make unrealistic assumptions for the sake of developing theory.

Weaknesses: I would consider the following to be possible weaknesses: a. The simulation does not add any value to the paper I think. It is a simple simulation, and we already have theoretical proofs for the algorithm. b. I am not sure whether the proof of inferiority of OMD w.r.t FTRL adds any substantive to the paper. It is already shown that adaptive MW algorithm solves the problem, and it seemed that the authors introduced the OMD angle because it makes the paper technically heavier. This is not really a criticism, since the observation is interesting, but it seemed a little tangential to the problem on hand.

Correctness: I checked the proofs in the main paper and they seem correct. I did not check anything in the supplement though.

Clarity: Yes.

Relation to Prior Work: Yes.

Reproducibility: Yes

Additional Feedback:

[Author Response · NeurIPS 2020]

**To all:** Thank you for exceptionally thoughtful and helpful comments!

**Reviewer 2:**

• **Further motivation for pseudo-regret:** You are correct that we mostly assumed that the setting (with the pseudo-
regret as the performance metric) was already motivated in previous work, and indeed there is room for including
more motivation in our introduction—we will accommodate this in our final version. In short, one application domain
where corruptions are natural is content/ads recommendation: the presence of malicious users affects the feedback
signal received by the learning algorithm, but the objective one cares about is the performance of the system (measured
via pseudo regret) on the true population of non-malicious users.

• **Performance of FTL:** This is an excellent comment: FTL is indeed a very natural algorithm in the pure stochastic
setting, and it would be interesting to see how it performs in the mildly corrupted case. We will give this some thought
for the final version, and at the very least include a comment about it as you suggested.

• **Plots for FTRL:** We have inspected in depth the issue you are pointing out to (we do agree that something appears
wrong there), and it turns out that while there is no bug in the experiments, they do illustrate a rather non-intuitive
behaviour: recall that there is a trivial upper bound of $O(\Delta T)$ on the pseudo-regret, which kicks in once $C$ becomes
very large; the latter bound actually increases with the gap $\Delta$! This explains the artifact you mentioned, which indeed
takes place only at high levels of corruption $C$. At the same time, there is of course no contradiction to our upper
bounds. Many thanks for highlighting this—to avoid confusion, we will rework the plots in the more interesting
regime where this artifact is negligible (or at the very least carefully discuss this confusing behavior).

• **Minor glitches in proofs:** Thank you for carefully inspecting the proofs and spotting those! We will of course make
sure all are corrected for the final version.

**Reviewer 3:**

• **Corruption in the losses and not only in the feedback:** This is a fantastic point, on which we will remark in the
final version: a similar analysis can give for the same MW algorithm an upper bound of order $\sqrt{C/\Delta}$ with respect to
the corrupted losses, which is also tight for any value of $C$. (In this sense, MW enjoys a "best of all worlds" guarantee
for any corruption level.)

• **Additive $+\Delta$ term on page 13, line 387:** Note that the summation is changed from $p_{t+1,i}$ to be over $p_{t,i}$; to include
the last term of the original summation an additive $\Delta$ is required. We will elaborate more in the final version.

• **Where $p_{t,i}(\mu_i - \mu_{i^*}) \geq 0$ is being used:** In Eq. (10) we upper bound the summation over $t = t_0 + 1, \ldots, T$ by the
summation over $t = 1, \ldots, T$; this holds due to the fact that each term of the summation is non-negative.

**Reviewer 4:**

• **Practical impact of the result:** Our primary focus in this paper was indeed theoretical, and we do not claim the
results to have immediate practical consequences. However, we believe that the broader issue of statistical learning
under adversarial corruptions is highly relevant to practice, and that understanding the basic and fundamental questions
in this space is crucial before moving on to studying more complex settings.

• **Significance of the technical contribution:** It is true that parts of our development rely on existing techniques in
online learning (and we tried to be super transparent about the relationships to those in our writing). Granted, the
experts problem is an extremely well studied one and it is always possible to find similarities in the vast literature on
the subject. That said, note that our arguments differ from those of [32,33] (for the analogous MAB setting) in a
substantial way and rely on somewhat surprising properties of the classic Entropy regularization (e.g., the statement
of Lemma 7 is entirely new and was quite illuminating to us). These are crucial for obtaining sharp regret bounds,
which are logarithmic in $N$ and independent of $T$ (vs. linear in $N$, logarithmic in $T$ in the MAB case).

• **Relation to other "beyond worst-case" analyses:** Our discussion of related work in this context focused on prior
work within (online) learning. As you correctly remark, going beyond standard worst-case analysis is an active
research agenda relevant to many other fields, some of which are surveyed in the pointer you provided. We will do our
best to include some more broader context in the final version (but it is hard to do justice to the vast literature on that).

**Reviewer 5:**

• **Value of simulations and comparison between FTRL and OMD:** We partially agree with your view that these are
secondary results, and the main contribution of the paper being the analysis of the (FTRL variant of) MW in the
corrupted setting. On the other hand, we also think that the proven gap between FTRL and OMD in this setting is
quite surprising given the literature on these meta-algorithms, and the fact that this gap grows with the corruption
level is particularly insightful and directly related to the problem at hand. (See also the insightful comments made by
Reviewer 2 on this aspect, who actually found this a notable strength of the paper.)

[Meta-Review · NeurIPS 2020]

All reviewers support accept. One point that was discussed after the rebuttal period was the following: The work [24] by Mourtada and Gaïffas proves the same result when C=0 or more generally for the "adversarial with a gap" setting, so how much more work is needed to show the general case with C\neq 0 considered in this paper? Given only two very brief mentions of [24] in the paper, this is highly unclear to the readers. Note that in [32, 33], this generalization is almost trivial. While reviewers do not believe that this is also the case here, we do agree that much more discussions on this are needed. So please make this clearer in the final version, along with other minor issues pointed out in the reviews.